# Diverse stimuli engage different neutrophil extracellular trap pathways

Elaine F Kenny[1]*, Alf Herzig[1], Renate Krüger[2,3], Aaron Muth[4], Santanu Mondal[4], Paul R Thompson[4], Volker Brinkmann[5], Horst von Bernuth[2,3,6,7], Arturo Zychlinsky[1]

[1]Department of Cellular Microbiology, Max Planck Institute for Infection Biology, Berlin, Germany; [2]Department of Paediatric Pneumology and Immunology, Outpatient Clinic for Primary Immunodeficiencies, Charité Medical School, Berlin, Germany; [3]Sozialpädiatrisches Zentrum, Charité Medical School, Berlin, Germany; [4]Department of Biochemistry and Pharmacology, University of Massachusetts Medical School, Worcester, United States; [5]Microscopy Core Facility, Max Planck Institute for Infection Biology, Berlin, Germany; [6]Labor Berlin, Section for Immunology, Charité–Vivantes GmbH, Berlin, Germany; [7]Berlin Centre for Regenerative Therapies, Charité Medical School, Berlin, Germany

**Abstract** Neutrophils release neutrophil extracellular traps (NETs) which ensnare pathogens and have pathogenic functions in diverse diseases. We examined the NETosis pathways induced by five stimuli; PMA, the calcium ionophore A23187, nigericin, *Candida albicans* and Group B Streptococcus. We studied NET production in neutrophils from healthy donors with inhibitors of molecules crucial to PMA-induced NETs including protein kinase C, calcium, reactive oxygen species, the enzymes myeloperoxidase (MPO) and neutrophil elastase. Additionally, neutrophils from chronic granulomatous disease patients, carrying mutations in the NADPH oxidase complex or a MPO-deficient patient were examined. We show that PMA, *C. albicans* and GBS use a related pathway for NET induction, whereas ionophores require an alternative pathway but that NETs produced by all stimuli are proteolytically active, kill bacteria and composed mainly of chromosomal DNA. Thus, we demonstrate that NETosis occurs through several signalling mechanisms, suggesting that extrusion of NETs is important in host defence.

*For correspondence: kenny@mpiib-berlin.mpg.de

## Introduction

Neutrophils are the most abundant white blood cell in the circulation and serve as the first line of host defence against pathogen attack. They are terminally differentiated, short lived cells that emerge from the bone marrow ready to react to the presence of pathogens (*Amulic et al., 2012*; *Kolaczkowska and Kubes, 2013*). Once a foreign molecule or endogenous threat is identified the neutrophil has a battery of mechanisms it can deploy to insure optimum removal of the hazard. These include the ability to phagocytose, degranulate and produce reactive oxygen species (ROS). The neutrophil can also produce chemokines and cytokines to alert other cells in the vicinity to the danger and thus maximise the host's immune response (*Scapini and Cassatella, 2014*).

Another form of defence utilised by the neutrophil is the release of decondensed chromatin decorated with antimicrobial peptides that can capture the pathogen in a process termed neutrophil extracellular trap (NET) formation (*Brinkmann et al., 2004*). NETosis has been primarily examined in response to phorbol 12-myristate 13-acetate (PMA), a potent mitogen and a robust NET inducer. Neutrophils also initiate NETosis in response to microbial infections and, similarly to PMA, these

**eLife digest** The immune system protects the body against microorganisms that can cause infections and diseases. Neutrophils are a type of immune cell that patrol the blood in search of germs. Once they encounter potentially harmful microbes, neutrophils eradicate them in different ways. One way to catch the germs is by using 'neutrophil extracellular traps', or NETs for short, to confine and kill the invaders.

NETs are web-like structures made up of anti-microbial proteins and the neutrophil's own DNA. The process of making NETs kills the neutrophil, as it eventually explodes to release the NETs. NETs play a key role in disease prevention, but producing too many NETs or producing them at the wrong time or in the wrong place can actually make certain diseases worse. Therefore, it is important to fully understand the signaling pathways and molecules the neutrophils use to make NETs.

Kenny et al. exposed neutrophils from healthy people to five different compounds known to cause the cells to make NETs, including some harmful molecules, a fungus and a bacterium. Microscopy was then used to count how many neutrophils made NETs in response to each of the five stimuli. Further experiments showed that neutrophils from patients with an immune system disorder produced fewer NETs when stimulated with some of the compounds, while the other stimuli caused neutrophils to produce the same levels of NETs as healthy individuals.

Kenny et al. also revealed that neutrophils use several different ways to produce and release NETs, depending on the stimulus used. Some of the ways required reactive oxygen species, such as hydrogen peroxide and enzymes, while others produced NETs without the need for these molecules. Lastly, Kenny et al. showed that the way the cells die after the NET is released is unique from other pathways that are known to kill cells.

Future work will aim to identify a single molecule that can block neutrophils from releasing NETs at the wrong time and place, without affecting the important role NETs play in fighting germs. Such a molecule could be developed into a drug for people with diseases like lupus or rheumatoid arthritis, where the release of NETs makes the disease worse not better.

activate protein kinase C (PKC), which in turn leads to calcium fluxes within the cell and activation of the NADPH oxidase signalling cascade resulting in the production of reactive oxygen species (ROS) (*Hakkim et al., 2011*; *Kaplan and Radic, 2012*). The hydrogen peroxide ($H_2O_2$) produced is in turn consumed by myeloperoxidase (MPO) to produce hypochlorous acid as well as other oxidants (*Papayannopoulos et al., 2010*). The production of ROS is responsible for the activation of the azurosome, a protein complex composed of MPO, the serine protease neutrophil elastase (NE) and cathepsin G among other granular proteins. The generation of oxidants by MPO liberates NE from the azurosome, allowing it to translocate to the nucleus where it aids in the decondensation of the chromatin by proteolyzing histones (*Metzler et al., 2014*). Finally, the cytoplasmic milieu mixes with the nuclear material as the nuclear and subsequently the plasma membrane break down, resulting in release of the NET.

This study describes the different pathways leading to NETs by comparing the induction of NETosis by several stimuli. Primary neutrophils from healthy donors were treated with five stimuli: (1) PMA, (2) the calcium ionophore A23187, (3) the bacterial toxin nigericin that acts as a potassium ionophore, (4) the dimorphic fungus *Candida albicans* and (5) the gram-positive bacteria Group B Streptococcus (GBS) and examined for the production of NETs. We tested a range of inhibitors against proteins involved in NETosis to clarify the essential elements in NET induction.

To study the role of ROS in NETosis, we tested neutrophils isolated from chronic granulomatous disease (CGD) patients. These patients have mutations in genes coding for subunits of the NADPH oxidase complex and as such their neutrophils cannot make ROS (*Heyworth et al., 2003*). Thus, these patients are highly susceptible to bacterial and fungal infections. We also tested neutrophils from a patient with a mutation in MPO.

Citrullination is a post-transcriptional modification resulting in the conversion of arginine to citrulline and is catalysed by a group of calcium-dependent proteins known as peptidylarginine

deiminases (PADs) (*Fuhrmann et al., 2015*). Recent studies have shown that citrullination occurs during NETosis (*Lewis et al., 2015*). We therefore also investigated if histone H3 is citrullinated during the induction of NETosis in response to the different stimuli.

Finally, we showed that the NETs generated by the five stimuli have similar properties and that NETosis is a unique form of cell death, different from classical cell death pathways involving apoptosis and necroptosis.

## Results

### A wide range of stimuli induce neutrophil extracellular traps (NETs)

We selected five representative and well-described NET inducers that are effective over a 2.5–4 hr time period: (1) PMA, (2) the calcium ionophore A23187 which is produced during the growth of *Streptomyces chartreusensis*, (3) the potassium ionophore nigericin which is derived from the bacteria *Streptomyces hygroscopicus*, (4) *Candida albicans* hyphae and (5) *Streptococcus agalactiae* or Group B streptococcus (GBS) and examined NETosis (*Figure 1A*). We visualised and quantified NETs in samples that were fixed and stained with antibodies directed against a complex of histone 2A, histone 2B and chromatin (*Losman et al., 1992*) and against neutrophil elastase (NE). Finally, the DNA was stained with the DNA-intercalating dye Hoechst 33342. We used the DNA stain to count the total number of neutrophils and NETs were quantified based on the presence of extracellular chromatin and a size exclusion protocol previously described (*Brinkmann et al., 2012*). Activating neutrophils with each of the stimuli resulted in a similar NET structure containing extracellular DNA co-localised with NE and chromatin (*Figure 1B–G*).

*Figure 1—figure supplement 1* shows that PMA (**B**), A23187 (**C**) and nigericin (**D**) produced NETs with similar kinetics over a 3–4 hr time course. *C. albicans* (**E**) and GBS (**F**) induced a slower rate of NETosis and non-stimulated cells remained NET free for the duration of the experiment.

*Videos 1–6* also visualise the induction of NETosis in response to the aforementioned stimuli over a 6 hr time course. All stimuli resulted in the release of extracellular DNA; however, the temporal order of nucleus decondensation and plasma membrane rupture was varied.

### PKC is required for PMA, *C. albicans* and GBS induced NETosis. Only PMA and nigericin require calcium to make NETs

PMA is a direct protein kinase C (PKC) activator which, in turn, leads to calcium fluxes within the cell and both of these processes are required for PMA-induced NETosis (*Gupta et al., 2014*; *Fuchs et al., 2007*). As anticipated, PMA-induced NET formation was blocked by the PKC inhibitor Gö6976 (*Figure 2A*) (*Gray et al., 2013*). *C. albicans* and GBS NET induction was also blocked by the PKC inhibitor, albeit to a lesser degree (*Figure 2C*). The two ionophores, conversely, did not require PKC (*Figure 2B*).

PMA (*Figure 2D*) and nigericin (*Figure 2E*), induced NETosis was impaired by the calcium chelator BAPTA-AM. This chelator reduced NET formation only slightly in response to A23187 (*Figure 2E*). Early work on neutrophil signalling revealed that ionomycin can induce a massive influx of calcium into the neutrophil, reaching a concentration greater than 1 µM (*Gennaro et al., 1984*). This abundance of intracellular calcium may have overwhelmed the ability of the BAPTA-AM to chelate the calcium at the concentration used. Pre-treatment of the neutrophils with higher concentrations of the calcium chelator resulted in spontaneous NET formation (data not shown) therefore making the A23187 calcium requirements difficult to assess. A previous study demonstrated a role for calcium in NETosis, however, suggesting that A23187 does in fact require the calcium flux it induces to produce NETs (*Gupta et al., 2014*). Notably, calcium chelation did not impair NETosis in response to *C. albicans* or GBS (*Figure 2F*).

Finally, as previously shown for PMA and *C. albicans*, A23187, nigericin and GBS NET induction is independent of transcription (*Figure 2G–I*) (*Sollberger et al., 2016*).

### Differential ROS requirements of NETs

Generation of reactive oxygen species (ROS) is a hallmark of PMA-induced NETosis. *Figure 3A* confirms that PMA induced a ROS burst in primary human neutrophils, peaking after 20 min of stimulation. This ROS burst was largely abolished by pre-treating the neutrophils with the ROS scavenger

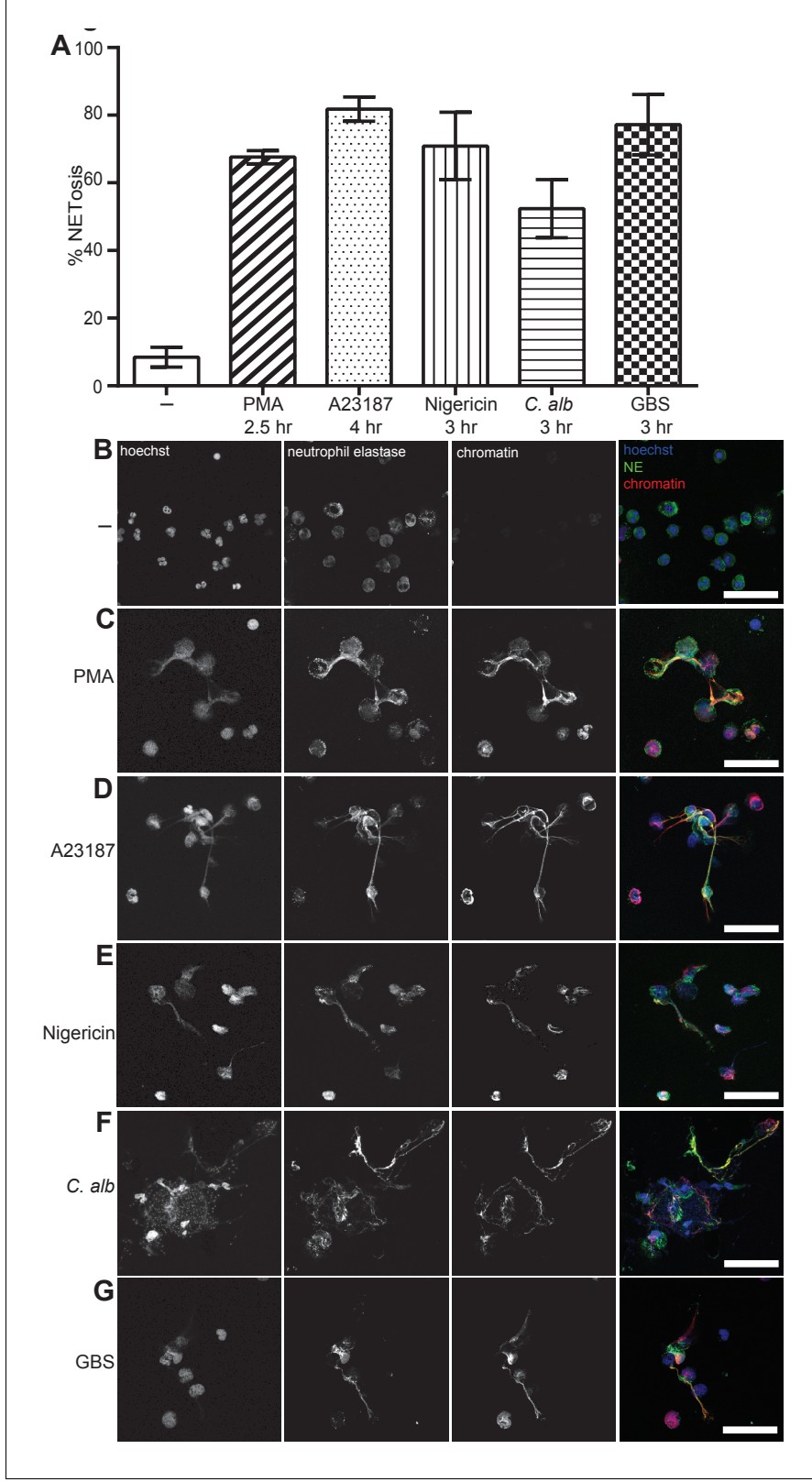

**Figure 1.** NETosis induction by a range of stimuli. Primary human neutrophils were stimulated for the indicated times with 50 nM PMA, 5 µM A23187, 15 µM nigericin, MOI 5 opsonized *C. albicans* or MOI 10 opsonized group B streptococcus (GBS), fixed with 2% PFA and incubated with a DNA stain (Hoechst) and immunolabeled with antibodies directed against Neutrophil Elastase (NE) and chromatin (**A–G**). (**A**) NETosis rate was quantified by

*Figure 1 continued on next page*

*Figure 1 continued*

immunofluorescence. Graph shows mean ± SEM from independent experiments with three different donors. (B–G) Representative confocal microscopy images of (B) non-stimulated neutrophils (-) or NETs induced by (C) PMA (D) A23187, (E) nigericin (F) *C. albicans* or (G) GBS and stained with Hoechst (blue) and immunolabeled for NE (green) and chromatin (red). Scale bars, 50 µm.

The following source data and figure supplement are available for figure 1:

**Source data 1.** This data is the mean values of three independent NETosis assays in response to the five stimuli of interest and was used to generate the histogram in *Figure 1A*.

**Figure supplement 1.** NET induction over time with the five stimuli of interest.

---

pyrocatechol. A23187 (*Figure 3B*) also induced a ROS burst, although with slower kinetics than PMA, peaking around 80 min post stimulation. Pyrocatechol also prevented this ROS burst. In contrast, nigericin did not induce any ROS production (*Figure 3B*). Opsonized *C. albicans* generated ROS (*Figure 3C*) peaked, like PMA 20 min after activation. GBS-induced ROS production to similar levels but with slower kinetics. ROS release by both microbes was abrogated by pyrocatechol. PMA, A23187, *C. albicans* and GBS induced ROS returned to basal levels 2 hr post-stimulation.

To test whether ROS were required for NETosis, we pre-incubated neutrophils with pyrocatechol before stimulation. As expected, ROS was absolutely required for PMA-induced NETosis (*Figure 3D*). Conversely, pyrocatechol did not affect the level of NETosis in response to A23187 or nigericin (*Figure 3E*). Interestingly, *C. albicans*-induced NETosis was impaired in the presence of the ROS scavenger, but GBS-induced NETosis was not (*Figure 3F*).

To confirm the ROS requirements for NETosis we isolated neutrophils from five patients with chronic granulomatous disease (CGD, mutations outlined in *Table 1*). As expected, the neutrophils from these patients were deficient in ROS production (*Figure 3—figure supplement 1A*). As previously described (*Fuchs et al., 2007*), CGD patient neutrophils did not undergo NETosis in response to PMA (*Figure 3G*) and were also significantly impaired in NET production in the presence of *S. aureus* (*Figure 3—figure supplement 1B*).

Notably, and confirming our data with the ROS scavenger, neither A23187 nor nigericin required ROS to generate NETs (*Figure 3H*). Intriguingly, and in contrast to data obtained with the ROS scavenger, there was no significant difference in the levels of NETosis comparing healthy vs. CGD patient neutrophils in response to *C. albicans*. This discrepancy was explained by the fact that *C. albicans* can induce a ROS burst itself in the absence of neutrophils and this was inhibited in the presence of the ROS scavenger pyrocatechol (*Figure 3—figure supplement 1C*). Indeed, we confirmed that *C. albicans* produces sufficient ROS to allow NETosis. We pre-incubated the fungus with pyrocatechol (*Figure 3—figure supplement 1D*) and showed that scavenging *C. albicans*-derived ROS also inhibited NET production. Moreover, by inhibiting ROS in both the neutrophils and the fungus NETosis was inhibited to a greater extent. Thus, the amount of ROS produced by the *C. albicans* was sufficient to allow NETs induction in CGD neutrophils. The amount of NETs were, however, decreased in CGD neutrophils infected with GBS

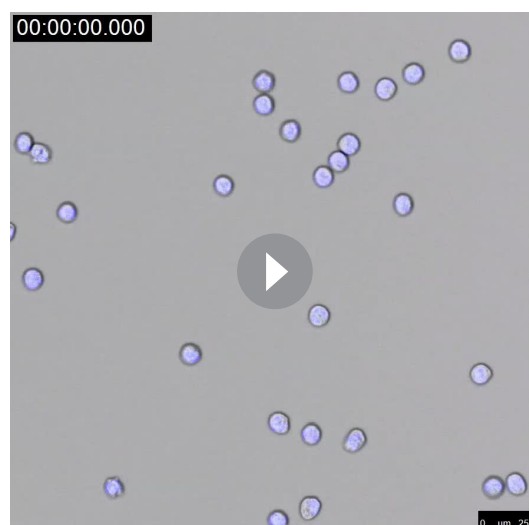

**Video 1.** No NETosis in non-stimulated primary neutrophils Primary neutrophils were stained with Draq5 (blue) and cell impermeable Sytox Green (green) and imaged for 6 hr using a Leica SP8 AOBS confocal microscope. Video is representative of three independent experiments.

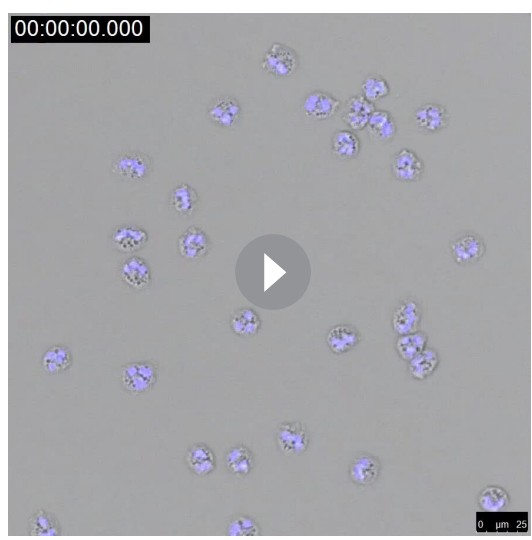

**Video 2.** PMA induced NETosis in primary neutrophils Primary neutrophils were stained with Draq5 (blue) and cell impermeable Sytox Green (green), stimulated with 50 nM PMA and imaged for 6 hr using a Leica SP8 AOBS confocal microscope. The appearance of the green colour indicated NETosis. Video is representative of three independent experiments.

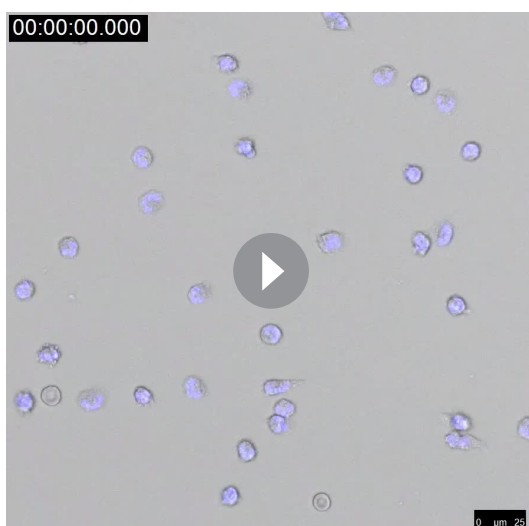

**Video 3.** A23187 induced NETosis in primary neutrophils Primary neutrophils were stained with Draq5 (blue) and cell impermeable Sytox Green (green), stimulated with 5 μM A23187 and imaged for 6 hr using a Leica SP8 AOBS confocal microscope. The appearance of the green colour indicated NETosis. Video is representative of three independent experiments.

(*Figure 3I*) when compared with cells isolated from healthy donors.

Overall, these data show that ROS generated by the NADPH oxidase, while absolutely essential for PMA-induced NETosis, are not necessary for NET production in response to both ionophores and only partially required for *C. albicans* and GBS-induced NETosis.

## Differential myeloperoxidase requirements of NETs

We explored whether myeloperoxidase (MPO) is also differentially required by the different stimuli. Aminobenzoic acid hydrazide (ABAH) is a potent and irreversible small molecule inhibitor of MPO. Pre-treatment of neutrophils with ABAH did not induce spontaneous NETosis and, as anticipated, significantly decreased PMA-induced NET formation (*Figure 4A*). This was confirmed with neutrophils isolated from a MPO-deficient patient stimulated with PMA (*Figure 4D*). Similar to the lack of ROS requirement in NETosis in response to ionophores, the MPO inhibitor did not affect NET production by A23187 or nigericin (*Figure 4B*) and neutrophils from a MPO-deficient individual underwent NETosis in response to both stimuli (*Figure 4E*). Interestingly, and contrary to the subtle role of ROS in NETosis induction, NETs induced by *C. albicans* or GBS stimulation required MPO (*Figure 4C and F*).

## Downstream of ROS and MPO, neutrophil elastase is differentially required for NETosis

Pre-treatment of healthy neutrophils with a highly specific small molecule NE inhibitor (*Macdonald et al., 2001*) did not result in spontaneous NETosis and significantly impaired PMA, *C. albicans* and GBS induced NETs (*Figure 5A and C*). NETosis in response to A23187 and nigericin did not require NE (*Figure 5B*). These data show that ionophores do not require the molecules relevant in other forms of NET induction such as ROS, MPO and NE.

## Histone citrullination occurs but is not required for NET induction

Stimulation of neutrophils with A23187, nigericin, *C. albicans* and GBS, but not PMA, resulted in Histone H3 citrullination (cit-H3) within 90 min as shown by Western blot analysis, (*Figure 6A*) confirming previous publications for both PMA and the calcium ionophore (*Neeli and Radic,*

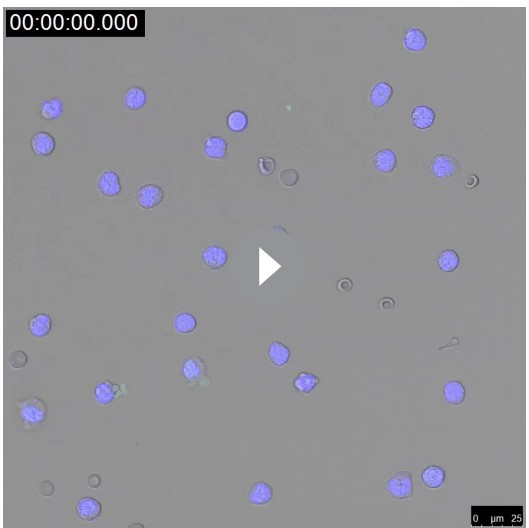

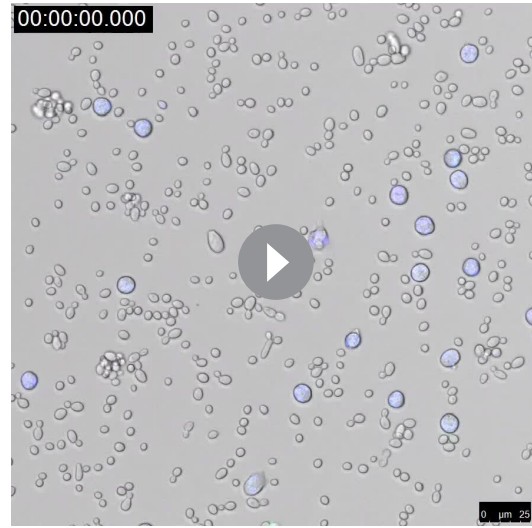

**Video 4.** Nigericin induced NETosis in primary neutrophils Primary neutrophils were stained with Draq5 (blue) and cell impermeable Sytox Green (green), stimulated with 15 µM nigericin and imaged for 6 hr using a Leica SP8 AOBS confocal microscope. The appearance of the green colour indicated NETosis. Video is representative of three independent experiments.

**Video 5.** *C. albicans* induced NETosis in primary neutrophils. Primary neutrophils were stained with Draq5 (blue) and cell impermeable Sytox Green (green), stimulated with MOI 5 *C. albicans* and imaged for 6 hr using a Leica SP8 AOBS confocal microscope. The appearance of the green colour indicated NETosis. Video is representative of three independent experiments.

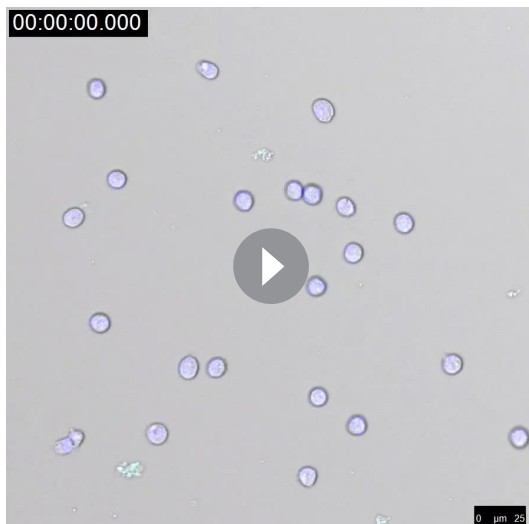

**Video 6.** GBS induced NETosis in primary neutrophils. Primary neutrophils were stained with Draq5 (blue) and cell impermeable Sytox Green (green), stimulated with MOI 10 GBS and imaged for 6 hr using a Leica SP8 AOBS confocal microscope. The appearance of the green colour indicated NETosis. Video is representative of three independent experiments.

*2013*). The citrullination data were confirmed by quantifying NETosis and the number of cit-H3-positive cells by immunofluorescence. Very few of the PMA induced NETs (*Figure 6B*) stained positively for cit-H3. Each of the other stimuli induced both NETosis and citrullination to varying levels (*Figure 6C and D*), confirming the data seen by Western blot analysis.

We next explored if citrullination was required for NET formation. Neutrophils were pre-treated with three inhibitors of PAD proteins: Cl-amidine and BB-Cl-amidine, both of which inhibit PAD2 and PAD4, and Thr-Asp-F-amidine (TDFA), a potent specific PAD4 inhibitor. Treating neutrophils with Cl-amidine, BB-Cl-amidine or TDFA did not induce NETosis spontaneously (*Figure 6E*). PMA induced NETosis was not affected by these inhibitors, consistent with the data obtained with GSK199, another PAD4-selective inhibitor (*Figure 6E*) (*Lewis et al., 2015*). Importantly, in response to A23187 or nigericin stimulation, NETosis remained intact after incubation with the three inhibitors (*Figure 6F*). Similarly, these inhibitors did not affect *C. albicans* or GBS-induced NETosis (*Figure 6G*). These data are the combination of 10 independent experiments with different

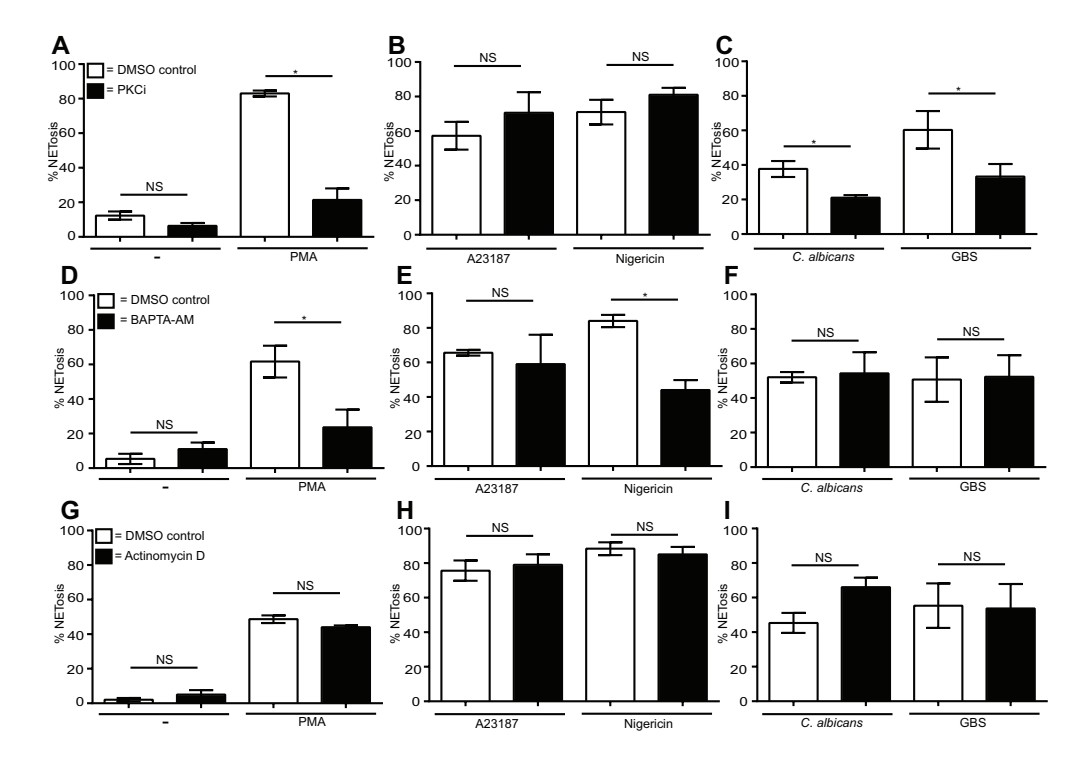

**Figure 2.** Differential requirements for PKC and calcium and a lack of requirement of transcription for NET induction by the stimuli of interest. (**A–C**) NETosis rate in PKC inhibited neutrophils. Primary neutrophils were pre-treated with the PKC inhibitor Gö6976 (1 μM) for 30 min and stimulated with (**A**) PMA, (**B**) A23187 or nigericin, and (**C**) *C. albicans* or GBS for 2.5–4 hr and analysed for NET production by immunofluorescence. (**D–F**) NETosis rate in neutrophils pre-treated with the calcium chelator BAPTA-AM (10 μM) for 30 min and stimulated with (**D**) PMA, (**E**) A23187 or nigericin and (**F**) *C. albicans* or GBS for 2.5–4 hr and analysed for NET production by immunofluorescence. (**G–I**) NETosis rate in neutrophils pre-treated with actinomycin D (1 μg/ml) for 30 min and stimulated with (**G**) PMA, (**H**) A23187 or nigericin and (**I**) *C. albicans* or GBS for 2.5–4 hr and then analysed for NET production by immunofluorescence. Graphs show mean ± SEM from three independent experiments. *p<0.05, NS = not significant.

The following source data is available for figure 2:

**Source data 1.** This data is the mean values of three independent NETosis assays in response to the five stimuli of interest in the presence of the PKC inhibitor Gö6976 (*Figure 2A–C*), the calcium chelator BAPTA-AM (*Figure 2D–F*) and actinomycin D (*Figure 2G–I*) and was used to generate the histograms in *Figure 2*.

donors. Each individual experiment is shown in *Figure 6—figure supplement 2*. The three inhibitors reduced citrullination in response to A23187, *C. albicans* and GBS, confirming that these inhibitors were active (*Figure 6—figure supplement 1A–C*).

These data show that histone H3 citrullination occurs during the course of NETosis in response to A23187, nigericin, *C. albicans* and GBS but not PMA-induced NETs. Moreover, the inhibitor assays demonstrate that PAD2 and PAD4 are not required for NETosis.

## All stimuli induce NETs that are proteolytically active, kill bacteria and are composed primarily of nuclear DNA

We next examined the properties of the NETs generated by the different stimuli. We began by examining the proteolytic activity of the NETs. As previously shown for PMA (*Papayannopoulos et al., 2010*), the induction of NETosis with all five stimuli resulted in the degradation of histone H3 at both 90 and 180 min (*Figure 7A*). Stimulation with PMA and nigericin resulted in strong degradation at the 90 min time point and a total loss of the full length histone H3 at 180 min. Conversely, A23187, *C. albicans* and GBS stimulation led to less degradation overall. This was further confirmed by assaying the degradation of FITC-labelled casein in the presence of NETs isolated from healthy neutrophils treated with the five stimuli (*Figure 7B*). NETs from all five

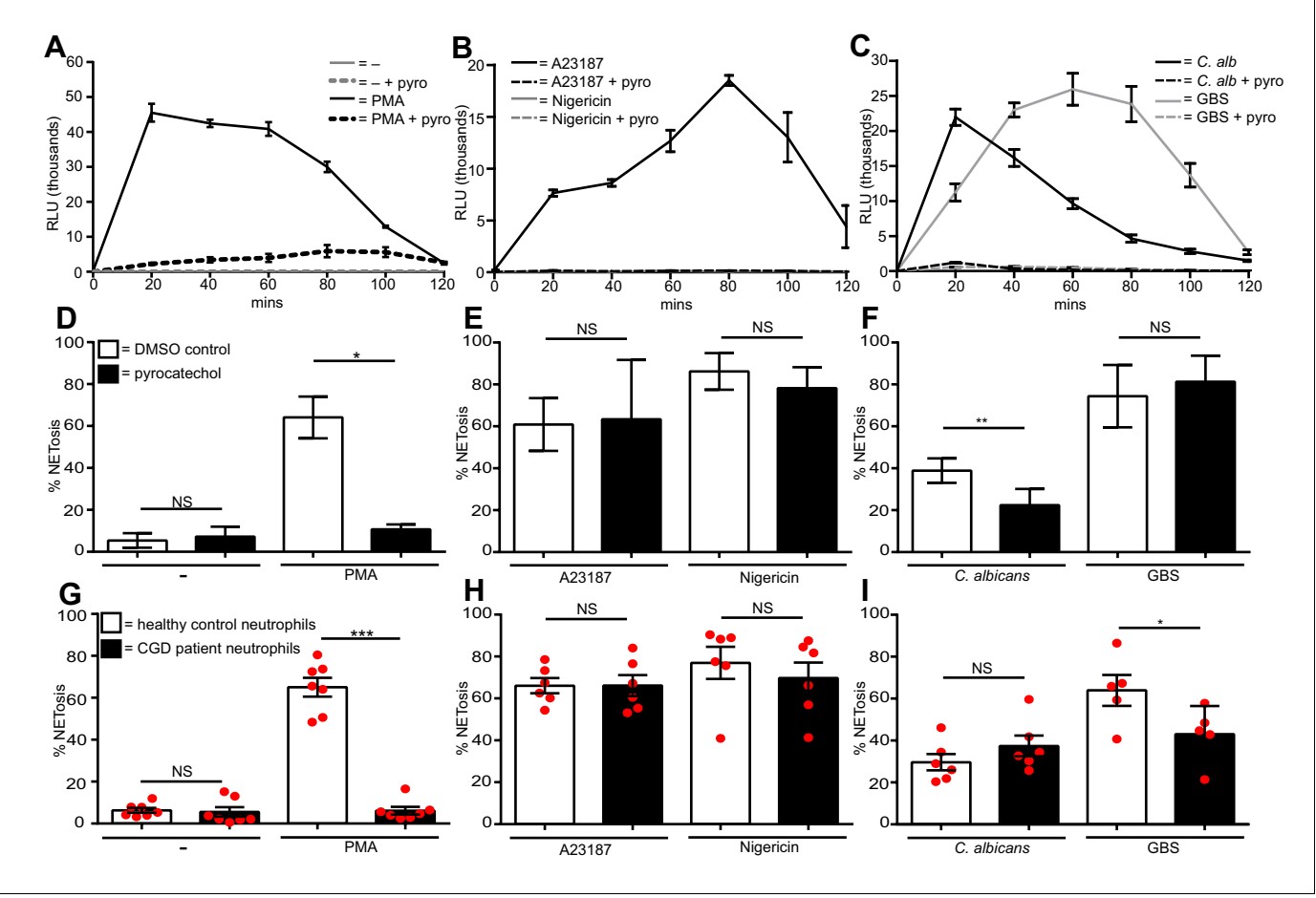

**Figure 3.** Diverse stimuli have different ROS requirements for NETosis. ROS production by neutrophils (A–C). ROS production was measured over a 2-hr time course in the presence or absence of the ROS scavenger pyrocatechol (pyro, 30 µM) in response to (A) PMA, (B) A23187 or nigericin and (C) *C. albicans* or GBS stimulation. Shown is a representative of three independent experiments. (D–F) NETosis rate of neutrophils pre-treated for 30 min with pyrocatechol or (G–I) NETosis rate of healthy control neutrophils and CGD patients stimulated with (D and G) PMA, (E and H) A23187 or nigericin and (F and I) *C. albicans* or GBS. (A–C) Graphs show mean ± SD from a representative of three independent experiments. (D–F) Graphs shows mean ± SEM from three independent experiments. (G–I) Graphs show mean ± SEM from five to seven independent experiments using neutrophils from five independent CGD patients (each represented by a red circle). *p<0.05, **p<0.01, ***p<0.001, NS = not significant.

The following source data and figure supplement are available for figure 3:

**Source data 1.** This data is the mean values of three independent NETosis assays in response to the five stimuli of interest in the presence of the ROS scavenger pyrocatechol and was used to calculate the histograms in *Figure 3D–F*.

**Figure supplement 1.** No ROS production in CGD patient neutrophils, S. aureus requires ROS for NET production and *C. albicans* produces ROS.

stimuli were capable of degrading FITC-labelling casein to a similar level indicating they are proteolytically active. The supernatant of non-stimulated neutrophils was used to determine the background level of degradation and a known concentration of trypsin was added as a positive control.

Next we tested the ability of the NETs to kill *E.coli*. Healthy neutrophils were treated for 4 hr with the stimuli to induce NETosis, phagocytosis was blocked with the addition of Cytochalasin D and *E. coli* were added for 1 hr in the presence or absence of DNase 1 (to degrade the NETs). NETs produced by all five stimuli were capable of limited killing that was blocked in the presence of DNase 1 (*Figure 7C*). The NETs were sonicated post *E.coli* incubation to release bacteria potentially trapped in clumps and skewing the killing counts. This did not affect the bacterial counts indicating that the NETs were in fact killing the bacteria.

**Table 1.** CGD patient donors. Nomenclature for genotypes is according to den Dunnen and Antonarakis (*den Dunnen and Antonarakis, 2001*).

| Donor | Age | Nucleotide change | Mutation | Amino acid change | Residual activity |
|---|---|---|---|---|---|
| 1 | 24 | CYBB c.742dupA | insertion | p.Ile248AsnfsX36 | No |
| 2 | 25 | CYBB c.868C > T | nonsense | p.Arg290X | No |
| 3 | 18 | CYBB c.1421T > G | missense | p.Leu474Arg | No |
| 4 | 26 | CYBB c.868C > T | nonsense | p.Arg290X | No |
| 5 | 29 | CYBA c.371C > T | missense | p.Ala124Val | Yes |

We also examined the NETs for the presence of mitochondrial DNA as this is seen in response to autoimmune stimuli such as ribonucleoprotein immune complexes (*Lood et al., 2016*). NETs produced in response to PMA, nigericin, *C. albicans* and GBS contained around 10% mitochondrial DNA and A23187 NETs contained around 25% (*Figure 7D*). This is in line with previous work demonstrating that NETs are mainly generated from chromosomal DNA (*Lood et al., 2016*).

Taken together this data revealed that NETs produced in response to all stimuli tested can degrade proteins, kill bacteria and mostly contain nuclear DNA.

## NETosis is distinct from other forms of cell death

To conclude, we compared NETosis, apoptosis, necrosis and necroptosis in neutrophils. We treated neutrophils with a caspase-3 inhibitor to block apoptosis or necrostatin to prevent necroptosis and measured NET formation revealing that NETosis is not affected by either of these inhibitors (*Figure 8A–C*).

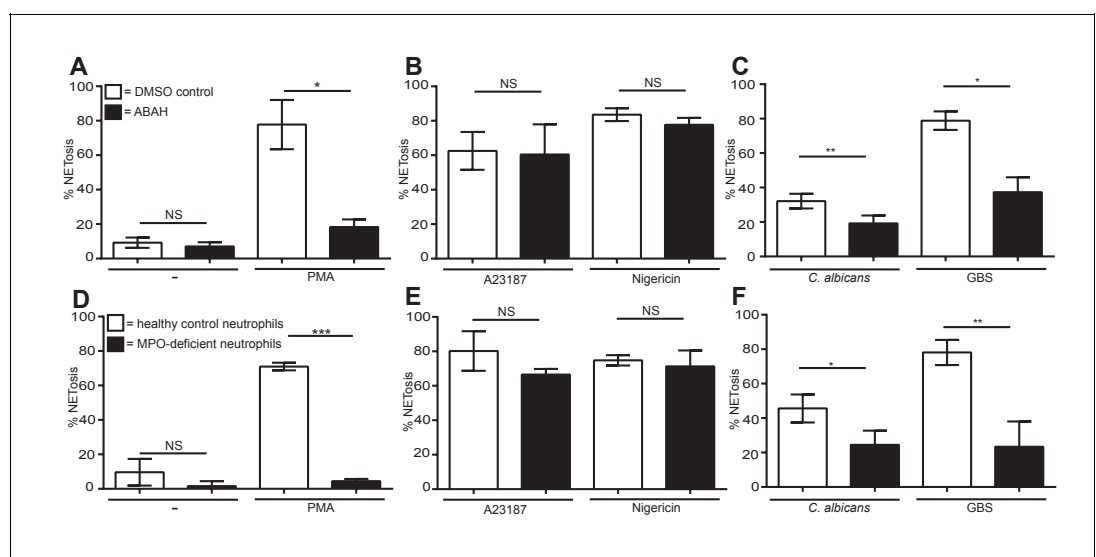

**Figure 4.** Myeloperoxidase is essential for PMA, *C. albicans* and GBS-induced NETosis but not for A23187 and nigericin-induced NET formation. (A–F) NETosis rate in response to (A and D) PMA, (B and E) A23187 or nigericin and (C and F) *C. albicans* or GBS. (A–C) Primary neutrophils were pre-treated for 30 min with 500 μM ABAH or a DMSO control, stimulated as indicated for 2.5–4 hr and analysed for NET production by immunofluorescence. Graphs show mean ± SEM from three independent experiments. (D–F) Healthy control neutrophils and neutrophils from a MPO-deficient patient were stimulated as outlined above. Graphs show mean ± SD from a representative of two independent experiments from a single MPO-deficient donor. *p<0.05, **p<0.01, ***p<0.001, NS = not significant.

The following source data is available for figure 4:

**Source data 1.** This data is the mean of three independent NETosis assays in response to the five stimuli of interest in the presence of the MPO inhibitor ABAH and was used to generate the histograms in *Figure 4A–C*.

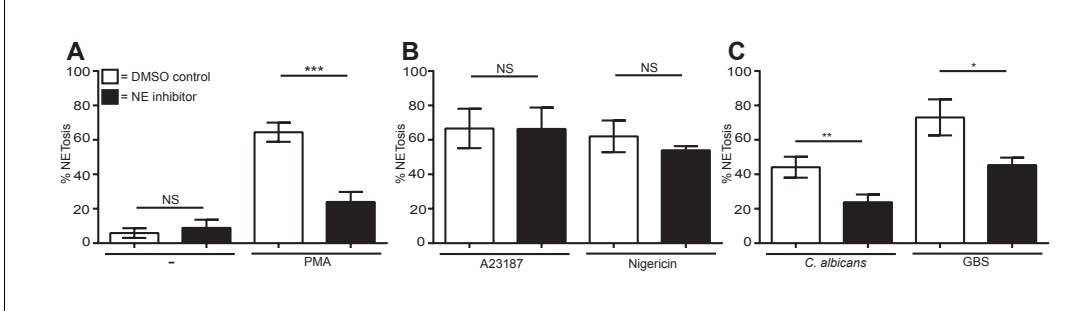

**Figure 5.** Neutrophil elastase is required for PMA, *C. albicans* and GBS-induced NETosis but not for A23187 or nigericin NET production. (A–C) NETosis rate of neutrophils during NE inhibition. Primary neutrophils were pre-treated for 30 min with a neutrophil elastase inhibitor (GW311616A, 20 μM) or a DMSO control and stimulated for 2.5–4 hr with (A) PMA, (B) A23187 or nigericin and (C) *C. albicans* or GBS and analysed for NET production by immunofluorescence. Graphs show mean ± SEM from three independent experiments. *p<0.05, **p<0.01, ***p<0.001, NS = not significant.

The following source data is available for figure 5:

**Source data 1.** This data is the mean of three independent NETosis assays in response to the five stimuli of interest in the presence of a NE inhibitor and was used to generate the histograms in *Figure 5A–C*.

Importantly, the apoptosis inducer staurosporine did not induce NET formation even after 6 hr incubation (*Figure 8D*). As a control, we showed that staurosporine-induced apoptosis in neutrophils as demonstrated by the presence of cleaved caspase-3. This cleavage was blocked by a caspase-3 inhibitor (*Figure 8—figure supplement 1A*). These data were confirmed with the pan-caspase inhibitor Z-VAD-FMK (data not shown). As expected, staurosporine did not induce LDH release (*Figure 8—figure supplement 1B*).

Finally, we induced necrosis with α-hemolysin from *Staphylococcus aureus* or necroptosis with a cocktail of TNF-α, Z-VAD-FMK and a SMAC mimetic or cycloheximide (CHX) and measured NETosis. Neither necrosis nor necroptosis activation induced NETosis beyond the level seen in the non-stimulated cells over a 6-hr time course (*Figure 8E*). LDH assays confirmed that α-hemolysin and the necroptosis cocktail-induced cell death (*Figure 8—figure supplement 1C and D*). As a control, we verified that necrostatin blocked cell death due to necroptosis (*Figure 8—figure supplement 1D*).

Lastly, we investigated whether LDH release occurs during NETosis. *Figure 8—figure supplement 1E* showed that PMA, nigericin, *C. albicans* and GBS stimulation resulted in LDH release greater than non-stimulated cells at 4 hr. A23187 treatment also resulted in LDH release but to a lesser extent.

In conclusion, these data show that NETosis is a unique form of cell death that does not utilise components of the pathways associated with apoptosis, necrosis or necroptosis.

## Discussion

Recent work focusing on the various stimuli that lead to NETosis has yielded contradictory results in response to the same stimuli. These discrepancies may arise from technical issues such as differences in neutrophil isolation protocols, the culture of neutrophils in different types of cell culture media and the concentration of stimuli used. With this in mind, we aimed here to confirm that our stimuli of interest generate NETs by use of the benchmark for NETosis analysis: quantifiable analysis along with fixed cell imaging and live cell videos.

Using these methods, we demonstrate here that NETs can be robustly induced by a broad range of stimuli, including PMA, the ionophores A23187 and nigericin and the more physiologically relevant stimuli *C. albicans* and GBS. Using neutrophils isolated from healthy donors and patients as well as small molecule inhibitors, we show that NET formation occurs through different signalling pathways. The NETs generated by all five stimuli were proteolytically active, kill bacteria and composed mainly of chromosomal DNA. We also show that NETosis is distinct from other cell death pathways such as apoptosis, necrosis and necroptosis.

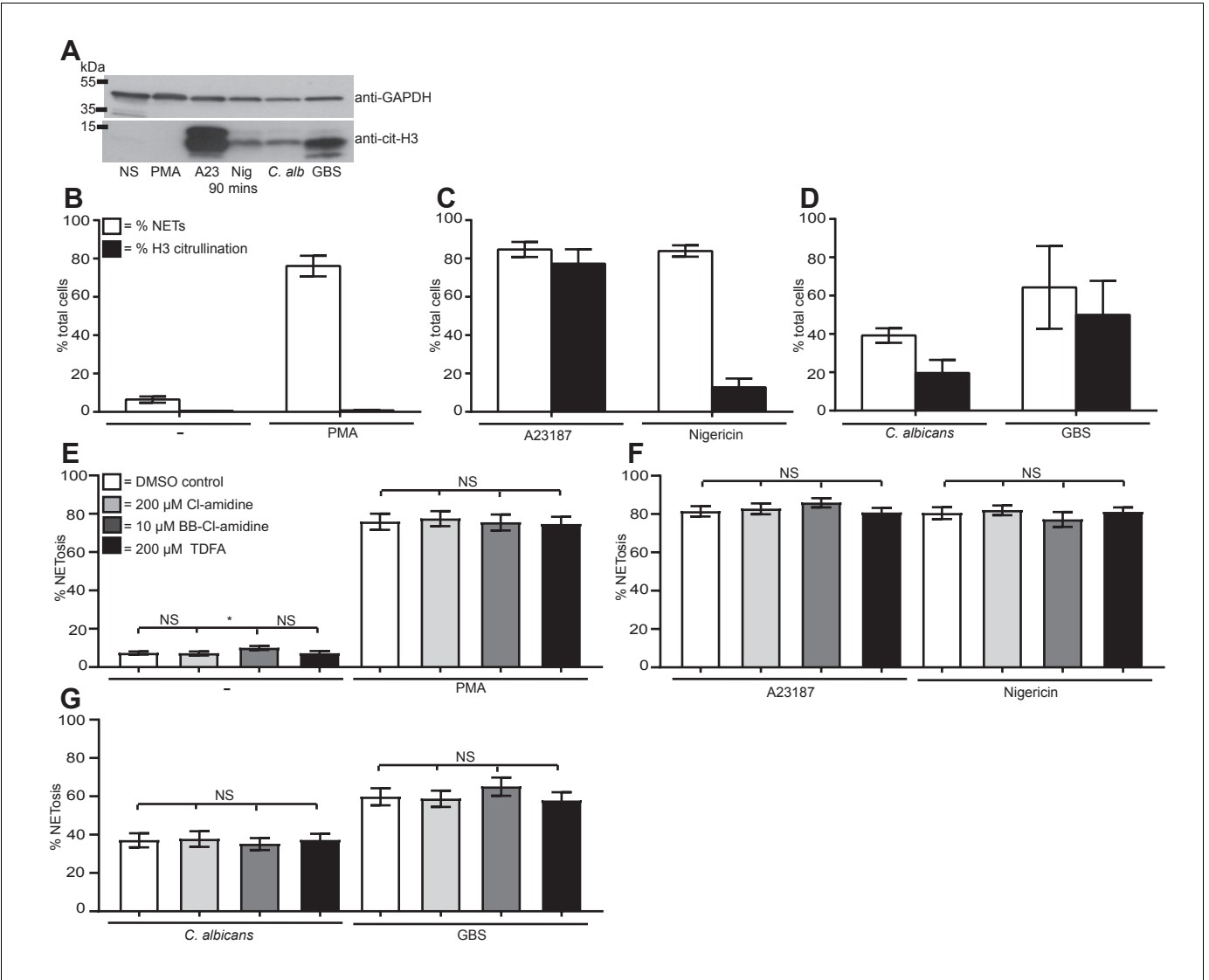

**Figure 6.** Citrullination of histone H3 occurs during NETosis but is not required for NET induction. (A–D) Histone H3 was citrullinated during NETosis in response to all stimuli bar PMA. (A) Primary neutrophils were stimulated for 90 min with PMA, A23187, nigericin, *C. albicans* or GBS, lysed and assayed for the presence of citrullinated histone H3 and GAPDH by SDS-PAGE electrophoresis and Western immunoblotting. (B–D) NETosis rate and percentage of citrullinated cells in response to (B) PMA, (C) A23187 or nigericin and (D) *C. albicans* or GBS. Graphs show mean ± SD from a representative of two independent experiments. (E–G) NETosis rate in neutrophils pre-treated with the PAD inhibitor Cl-amidine at 200 μM, BB-Cl-amidine at 10 μM, TDFA at 200 μM, or DMSO as control and stimulated with (E) PMA, (F) A23187 or nigericin and (G) *C. albicans* or GBS and analysed for NET production by immunofluorescence. Graphs show mean ± SEM from 10 independent experiments. *p<0.05, NS = not significant.

The following source data and figure supplements are available for figure 6:

**Source data 1.** This data is the mean of ten independent NETosis assays in response to the five stimuli of interest in the presence of the PAD inhibitors and was used to generate the histograms in *Figure 6*.

**Figure supplement 1.** PAD inhibitors reduce histone H3 citrullination.

**Figure supplement 2.** PAD inhibitors do not prevent NETosis.

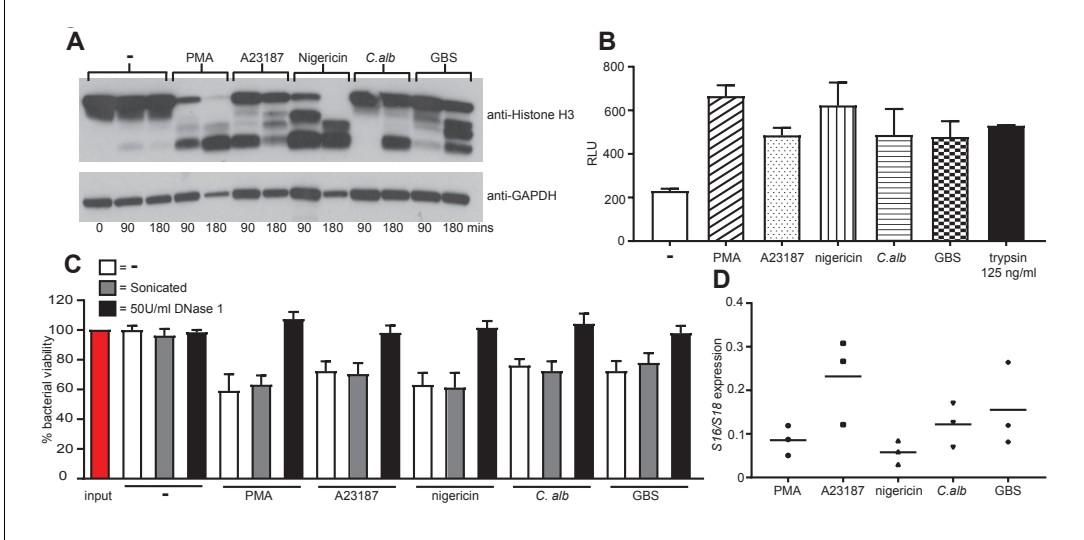

**Figure 7.** NETs are proteolytically active, kill bacteria and are mainly composed of chromosomal DNA. (**A**) NETosis leads to histone H3 degradation. Primary neutrophils were stimulated for 90 and 180 min with PMA, A23187, nigericin, *C. albicans* or GBS, lysed and assayed for the presence of histone H3 and GAPDH by SDS-PAGE electrophoresis and Western immunoblotting. Shown is a representative of three independent experiments. (**B**) Isolated NETs are proteolytically active. NETosis was induced for 4 hr, NETs were isolated after treatment with *Alu*I for 20 min, the DNA content was determined and 200 ng/ml DNA was tested for its proteolytic activity using the Pierce Fluorescent Protease Assay Kit according the manufacturer's instructions. 100 μl of non-stimulated neutrophil supernatant was used to determine the background activity and 125 ng/ml trypsin was added as a positive control. (**C**) NETs can kill *E. coli*. Neutrophils were stimulated to produce NETs for 4 hr. Phagocytosis was inhibited by the addition of Cytochalasin D and *E. coli* at a MOI of 1 were added in the presence or absence of 50 U/ml DNase 1. After 1 hr the cells, NETs and *E. coli* were collected (selected samples were sonicated), serially diluted, plated on tetracycline-resistant agar plates and incubated for 24 hr at 37°C followed by CFU counts to determine killing. (**D**) NETs are primarily composed of chromosomal DNA. 4 hr post NET induction the NETs were isolated by MNase treatment, followed by proteinase K treatment. NET DNA was isolated by phenol-chloroform extraction and the ratio of *S18* to *S16* DNA was analysed by real-time PCR. Graphs show mean ± SEM from three independent experiments.

The data shown here demonstrates that PMA-induced NETs require PKC, calcium, ROS, MPO and NE. Conversely, Histone H3 is not citrullinated upon stimulation with PMA and transcription plays no role as has been previously shown (*Sollberger et al., 2016*).

Clarifying the mechanisms of NET formation in response to *C. albicans* and GBS proved to be challenging, perhaps due to the need to culture fresh microbes for each experiment. Similar to PMA both microbes require PKC, MPO and NE and do not require transcriptional activation for NET formation. However, both microbes induce the citrullination of histone H3. Despite this PAD2/PAD4 activity is not required for NETosis in response to the microbes.

The role of ROS is less conclusive as there is a discrepancy between comparing healthy neutrophils treated with ROS scavengers and cells isolated from CGD patients. Indeed, *C. albicans* induces significantly less NETs when ROS were pharmacologically abrogated in neutrophils from healthy donors. In contrast, neutrophils from CGD patients produced similar amounts of NETs as those isolated from healthy controls. This suggests that the ROS used by *C. albicans* do not originate from the NADPH oxidase complex. We show here that indeed *C. albicans* itself can produce ROS thus circumventing the ROS requirements of the neutrophil by producing sufficient levels of ROS to allow CGD neutrophils to form NETs. These results are very much in line with the clinical phenotype in which patients with CGD suffer more frequently from invasive infections with *A. fumigatus* than with *C. albicans* (*Henriet et al., 2012*). Recently, a study demonstrated that CGD patient neutrophils produce significantly less NETs in response to *A. fumigatus* than healthy neutrophils further adding to our evidence that neutrophils from CGD patients react diversely to fungal infections (*Gazendam et al., 2016*).

The role of ROS in GBS-induced NETosis is confounded by the ability of healthy neutrophils to produce normal levels of NETs in the presence of the ROS scavenger but a significantly reduced level of NETosis in the CGD patient neutrophils. The CGD patient data suggest a requirement for

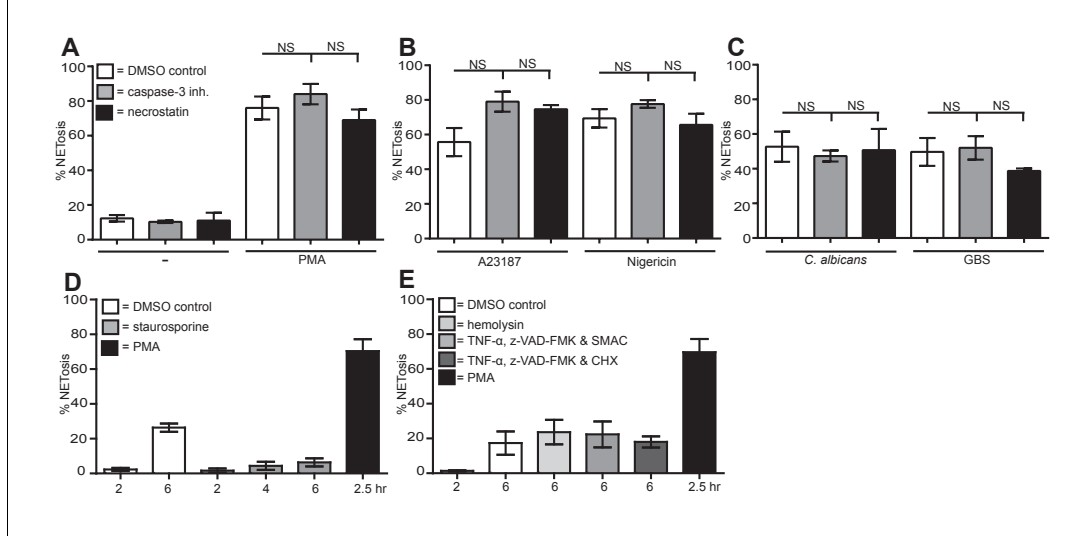

**Figure 8.** NETosis is a unique form of cell death different from apoptosis, necrosis and necroptosis. (A–C) NETosis occurs in the presence of apoptosis and necroptosis inhibitors. Primary human neutrophils were pre-treated for 30 min with 20 μM caspase-3 inhibitor or 30 μM necrostatin or a DMSO control and stimulated with (A) PMA, (B) A23187 or nigericin and (C) *C. albicans* or GBS for 2.5–4 hr and analysed for NET production by immunofluorescence. Graphs show mean ± SEM from three independent experiments. (D) NETosis rate in the presence of the apoptosis inducer staurosporine. Primary neutrophils were stimulated for 2–6 hr with staurosporine (500 nM) or PMA and analysed for NET induction by immunofluorescence. Graphs show mean ± SEM from three independent experiments. (E) NETosis rate in response to necrosis or necroptosis inducers. Primary neutrophils were stimulated with α-hemolysin (25 μg/ml) to induce necrosis or with TNF-α (50 ng/ml), Z-VAD-FMK (50 μM) and a SMAC mimetic (100 nM) or cycloheximide (25 μg/ml) to induce necroptosis for 6 hr and analysed for NET production by immunofluorescence. Graphs show mean ± SEM from three independent experiments. NS = not significant.

The following source data and figure supplement are available for figure 8:

**Source data 1.** This data is the mean of three independent NETosis assays in response to the five stimuli of interest in the presence of necrostatin or caspase 3 inhibitor and was used to generate the histograms in *Figure 8A–C*.

**Figure supplement 1.** Apoptosis, necrosis and necroptosis can be induced in primary neutrophils, NETosis results in LDH release.

NAPDH oxidase-dependent ROS for GBS induced NETs that is perhaps not revealed by the ROS scavenger. As seen in *Figure 3C*, there was some residual ROS production in response to GBS in the presence of the ROS scavenger. This level of ROS may be sufficient for the GBS to induce NETosis in a manner similar to *C. albicans* in which the amount of ROS produced by the fungus allows NETosis to occur in the CGD patient neutrophils. It must also be noted that although the ability of GBS to induce NETosis was significantly reduced in the CGD patients, a high level of NETosis still occurred in these cells. Taken together, these data suggest that while PMA absolutely relies on NADPH oxidase derived ROS for NETosis *C. albicans* and GBS can circumvent this need to some degree. This could be due to the ability of the microbes to generate ROS themselves that is then hijacked by the neutrophil to generate NETs in the absence of a self-source of ROS.

This is in contrast with NETosis induction by *S. aureus* which was dependent on the ability of the neutrophils to generate ROS as outlined in the original study on NETosis in CGD patient neutrophils (*Fuchs et al., 2007*). This suggests that NETosis induced by physiologically relevant stimuli is also very diverse in the signalling pathways utilised and hence challenging to clarify. Indeed, *Leishmania amazonensis* can induce NETosis in the absence of ROS production (*Rochael et al., 2015*; *DeSouza-Vieira et al., 2016*).

The role of ROS production in the generation of NETs is further confounded by recent work demonstrating that the mitochondria can also be a source of ROS in NETosis in response to calcium ionophores (*Douda et al., 2015*) or ribonucleoprotein-containing immune complexes (RNP ICs) (*Lood et al., 2016*). These data demonstrate that outside of the NADPH-oxidase complex the neutrophil has other ROS sources that are sufficient to induce NETosis.

Conversely, activation with the ionophores A23187 and nigericin does not require PKC, ROS, MPO or NE or transcriptional activation and calcium only has a limited role. Histone H3 is citrullinated upon activation by these stimuli; however, pre-treatment with PAD inhibitors does not affect the ability of the neutrophils to make NETs. This suggests that citrullination (of histone H3 at least) is a consequence of NETosis, and that PAD4 is not required for NET formation. Ionophore and PMA-induced NETosis appears to be distinct, at least in the few components of the signal transduction cascade already described.

A23187 is a calcium ionophore that causes a massive influx of calcium and nigericin stimulates potassium effluxes in cells which also results in the influx of calcium demonstrating the similarity between the ionophores in their mechanism of NET induction (*Yaron et al., 2015*). This flooding of the neutrophil with calcium ions could thus result in perturbation of the membrane potential and cell death that ultimately releases NETs. While the method to initiate NET induction by the ionophores is very different to that seen in response to PMA, the end product appears to be similar.

Auto-antibodies directed against citrullinated proteins are commonly found in the serum of rheumatoid arthritis patients (*van Venrooij et al., 2004*) and as such elucidating the origin of these modified proteins is of great therapeutic interest. Recent research suggests that PAD enzymes, in particular PAD4, are associated with the induction of NETosis. Consistent with this, PAD4-deficient mice do not generate NETs in response to a calcium ionophore which is in direct contrast to the data presented here (*Martinod et al., 2013*). However, the readout for NETosis used in this study was the presence of extracellular DNA decorated with citrullinated histone H3. Since deficiency in PAD4 results in no citrullination of histones this study lacks a readout in the PAD4-deficient cells that would confirm the presence or absence of NETs such as staining with antibodies against NE or MPO on the extracellular DNA. It is also important to note that these experiments were carried out using murine neutrophils which may not behave similarly to human cells (*Bardoel et al., 2014*). Importantly, studies examining the requirements of the PAD enzymes in human NETosis, using the same PAD inhibitors, also demonstrate a very limited inhibition of NETosis in response to a calcium ionophore and *S. aureus* (*Hosseinzadeh et al., 2016*; *Lewis et al., 2015*). Thus, it appears that while PAD enzymes might be important for murine neutrophils to generate NETs, this effect is not seen in human neutrophils.

One additional discrepancy between our data and published reports is the observation that PAD inhibitors (both selective and pan-PAD) show efficacy in multiple mouse models of SLE and RA (*Knight et al., 2015*; *Ghari et al., 2016*; *Kawalkowska et al., 2016*). These studies suggest that citrullination is important in disease pathogenesis and as such could affect NET function. We do not understand how NETs function as antimicrobials, immune cell activators or in coagulation. It is possible that these NET functions are altered by the citrullination of NETs components. Indeed, the phenotype observed in the PAD4-deficient mice could potentially be attributed to the effectiveness of NETs and not necessarily NET formation.

A recent review highlights the wide range of proteins and pathways required for NETosis in response to a variety of stimuli with emphasis on the questions surrounding the role of citrullination in NETosis (*Konig and Andrade, 2016*). It states that ionophores and bacterial pore-forming toxins induce a pathway within neutrophils that results in the citrullination of histones but that is distinct from NETosis. They term this form of neutrophil cell death leukotoxic hypercitrullination (LTH) and suggest it is not antimicrobial but a bacterial strategy to kill neutrophils. The data shown here demonstrates that in the presence of the calcium ionophore or nigericin nuclear decondensation occurs and results in the extrusion of DNA, chromatin and peptides from neutrophils, albeit in a different manner to that utilised by PMA, *C. albicans* and GBS. Whether these extruded DNA and proteins are antimicrobial, however, requires further investigation.

The different mechanisms of neutrophil cell death have been studied in detail and as such the data presented here can be included in the body of evidence that NETosis is in fact a distinct cell death mechanism utilised to aid in pathogen killing by neutrophils (*Fuchs et al., 2007*; *Remijsen et al., 2011b*, *2011a*). However, two recent studies on the role of necroptosis in NETs induction present contrasting evidence for and against the requirements of necroptosis (*Amini et al., 2016*; *Desai et al., 2016*). Our data strengthens the argument that necroptosis is a separate cell death signalling cascade that is not required by neutrophils to induce NETosis.

Our observations show that there are different paths to NETosis in human cells. The elucidation of these pathways is of importance due to the ancient nature of chromatin release as a form of host

defence as has been identified in both the animal and plant kingdoms. Therefore, it is unsurprising that NETosis is induced through a wide range of pathways (*Tran et al., 2016*).

Consequently, the clarification of these different pathways to NETosis has definite therapeutic relevance. There is a genuine need to identify NET inhibitors to alleviate or prevent many diseases including cystic fibrosis, thrombosis, malaria and sepsis (*Kaplan and Radic, 2012*; *Brinkmann and Zychlinsky, 2012*). NETs are present in the sputum of cystic fibrosis (CF) patients and contribute to the viscosity of the sputum (*Manzenreiter et al., 2012*). NETs are evident in the thrombus in deep vein thrombosis (DVT) and disease activity is reflective of NET markers in the plasma (*Fuchs et al., 2012*). Many autoimmune diseases such as Systemic lupus erythematosus (SLE) and vasculitis also show a very strong NET phenotype with regard the presence of autoantibodies against proteins readily released from neutrophils in the process of NETosis such as anti-dsDNA and anti-neutrophil cytoplasmic autoantibodies (*Hakkim et al., 2010*; *Kessenbrock et al., 2009*).

This study will aid in the development of tools to help combat the detrimental effects of NETosis while balancing this with the need for the neutrophils to fulfil their purpose in the presence of a pathogen and induce their unique cell death program.

## Materials and methods

### Inhibitors

Gö6976 (PKC, Biozol), BAPTA-AM (Thermo Fisher Scientific), Actinomycin D, GW311616A (NE) and pyrocatechol (Sigma-Aldrich), 4-Aminobenzoic acid hydrazide (ABAH, Cayman chemical), Cl-amidine (*Causey and Thompson, 2008*), BB-Cl-amidine (*Knight et al., 2015*), TDFA (*Jones et al., 2012*), caspase-3 inhibitor and necrostatin (Merck-Millipore).

### Donor consent

Blood samples were collected according to the Declaration of Helsinki with study participants providing written informed consent. All samples were collected with approval from the ethics committee–Charité –Universitätsmedizin Berlin. Healthy neutrophils were isolated from blood donated anonymously at the Charité Hospital Berlin.

### Strains and media

*Candida albicans* clinical isolate SC5314 was cultured overnight at 30°C in YPD media. GBS was grown on a 6% sheep blood agar plate overnight at 37°C, sub-cultured in Todd-Hewitt broth (Sigma-Aldrich) for 2–3 hr until the $OD_{600nm}$ reached 0.5. *Staphylococcus aureus* was prepared as previously described (*Fuchs et al., 2007*). *E. coli* XL1-Blue (Stratagene) was cultured overnight at 37°C in LB plus tetracycline. The *C. albicans*, GBS and *S.aureus* were opsonized for 30 min at 37°C with 10% human serum before addition to the neutrophils. This also ensured hyphal growth of the *C. albicans*.

### Neutrophil isolation and NET induction

Human neutrophils were isolated by centrifuging heparinized venous blood over Histopaque 1119 (Sigma-Aldrich) and subsequently over a discontinuous Percoll (Amersham Biosciences) gradient as previously described (*Fuchs et al., 2007*). Experiments were performed in RPMI-1640 (w/o phenol red) supplemented with 10 mM HEPES and 0.05% human serum albumin. Cells were seeded at $10^5$/ well (24-well plate) for NET experiments and stimulated with PMA, staurosporine, cycloheximide (Sigma-Aldrich), A23187 (Santa Cruz Biotechnology Inc.), Nigericin (InvivoGen), *Candida albicans* SC5314 hyphae, GBS, α-hemolysin (generated as previously described [*Virreira Winter et al., 2016*]), *Staphylococcus aureus* (prepared as previously described [*Fuchs et al., 2007*]), TNF-α (Thermo Fisher Scientific), z-VAD-FMK (Enzo) or SMAC mimetic (Birinapant, ChemieTek) for 2–6 hr. Where applicable, cells were pre-treated inhibitors for 30 min before stimulation.

### NET staining and quantification

Neutrophils seeded on glass coverslips were stained and quantified as previously described (*Brinkmann et al., 2012*). Briefly, cells were fixed in 2% paraformaldehyde (PFA) post-NET induction, permeabilized on 0.5% Triton-X100, blocked for 30 min in blocking buffer. Neutrophils were then

stained with the following primary antibodies: anti-neutrophil elastase (Calbiochem: 481001, RRID: AB_212213) and antibodies directed against the subnucleosomal complex of Histone 2A, Histone 2B, and chromatin ([Losman et al., 1992], generated in house). The secondary antibodies donkey anti-mouse Cy3 (Jackson ImmunoResearch Labs: 715-175-150, RRID:AB_2340819) and donkey anti-rabbit Alexa Fluor488 (Life Technologies: A11008, RRID:AB_143165) were used. Finally, the samples were stained with Hoechst 33342 (Immunochemistry: 639, RRID:AB_2651135) and mounted with Mowiol. Image acquisition was c using a Leica DMR upright fluorescence microscope equipped with a Jenoptic B/W digital microscope camera and analysed using ImageJ/FIJI software.

## Confocal microscopy

Primary human neutrophils ($10^6$) were washed once by centrifugation (300 g, 10 min, RT) in imaging medium (20 mM HEPES, 2.5 mM KCl, 1.8 mM CaCl2, 1 mM MgCl2, 0.1% Human Serum Albumin, pH 7.4) (Sigma Aldrich) and then resuspended in 4 ml imaging medium containing 2 μM draq5 (Biostatus) and 0.5 μM Sytox Green (Thermo Fischer Scientific). Each well of an eight-well ibidi treat dish (ibidi) was filled with 200 μl of that suspension and cells were allowed to settle down for 30 min at imaging temperature. NETs were induced by PMA, A23187, nigericin, *C. albicans* or GBS at the concentrations outlined above. Imaging was performed with a Leica SP8 AOBS confocal microscope equipped with a motorized stage and temperature-controlled chamber at 36°C. Images (2048*2048 pixels) were acquired at 0.5% maximal laser intensities every 2 min for each well for a total duration of 6 hr.

## ROS assay

Neutrophils were seeded at concentration of $1 \times 10^5$ cells per well in 200 μl RPMI (w/o phenol red) supplemented with 10 mM HEPES, 0.05% human serum albumin, 50 μM luminol and 1.2 units/ml horseradish peroxidase and pre-treated with pyrocatechol for 30 min at 37°C. The cells were then stimulated for 2 hr with the indicated stimuli and luminescence was measured over time in a VICTOR Light luminescence counter from Perkin Elmer.

## Western blot

Neutrophil lysates were generated from $5 \times 10^6$ cells 90, 180 min (cit-H3 and H3) or 3 hr (caspase-3) post stimulation by lysis in RIPA buffer (50 mM Tris-HCl Ph 8.0, 150 mM NaCl, 1 mM EDTA, 1% NP-40, 0.5% sodium deoxycholate, 0.1% SDS, 10 mM sodium fluoride, 25 mM sodium pyrophosphate) supplemented with protease inhibitor cocktail (Sigma-Aldrich), 20 μM neutrophil elastase V inhibitor and 20 μM cathepsin G inhibitor (219372, both Calbiochem). Protein lysates were quantified by bicinchoninic acid assay (BCA assay, Pierce) according to manufacturer's instructions. Protein lysates were resolved by sodium dodecyl sulfate–polyacrylamide gel electrophoresis followed by analysis via Western immunoblotting using an anti-citrullinated Histone H3 primary antibody (abcam: ab5103, RRID:AB_304752), an anti-histone H3 antibody (Active Motif: 39164, discontinued) an anti-cleaved Caspase-3 antibody (9661, RRID:AB_2341188), anti-$\beta$-actin (5057S, RRID:AB_10694076) or anti-GAPDH (all Cell Signaling Technology: 5014S RRID:AB_10693448) and anti-rabbit HRP (Jackson ImmunoResearch Labs: 111-035-144, RRID:AB_2307391).

## Protease activity assay

NETs were generated as described above using $1.5 \times 10^6$ cells/point. The NETs were isolated as previously described (*Barrientos et al., 2013*). Briefly, the samples were treated with 4 U/ml *Alu*I, the NETs were collected, their DNA was quantified using the Quant-iT PicoGreen dsDNA Assay Kit (Thermo Fischer Scientific) and the protease activity of 200 ng/ml of NET DNA was quantified using the Pierce Fluorescent Protease Assay kit according to the manufacturer's instructions. 100 μl of non-stimulated neutrophil supernatant was used to calculate background protease activity and 125 ng/ml trypsin was used as a positive control.

## Bacterial killing assay

NETs were generated as described above using $1 \times 10^6$ cells/point and stimulated for 4 hr. Bacterial killing was assayed as previously described (*Ermert et al., 2009*). Briefly, once NETosis was induced (visualised by light microscopy), the cells were treated with 10 μg/ml Cytochalasin D (Sigma-Aldrich)

for 15 min to block phagocytosis. A subset of samples were treated with DNase 1 at 50 U/ml to degrade the NETs prior to killing. A tetracycline-resistant *E. coli* strain was added to the neutrophils at a MOI of 1 and incubated at 37°C for 1 hr. The cells and *E. coli* were collected, a subset of samples were sonicated to release any trapped bacteria, serially diluted, plated on tetracycline-treated agar plates and incubated at 37°C for 24 hr. Finally CFUs were counted.

## Quantitative real-time PCR of mitochondrial and nuclear content of NETs

NETs were generated as described above using $1.5 \times 10^6$ cells/point for 4 hr. NETs were released, DNA was isolated and analysed for nuclear (*S18*) versus mitochondrial (*S16*) content by Q-PCR as previously described (*Lood et al., 2016*).

## LDH assay

Neutrophils were seeded at $1 \times 10^5$ cells/well in a 96-well plate and treated for 21 hr with the indicated stimuli. LDH release was quantified from the supernatants using Cytotox 96 Non-Radioactive Cytotoxicity Assay (Promega) according to the manufacturer's instructions.

## Statistics

Data are presented as mean ± SEM unless otherwise noted and were analysed using a two-sided Student t test. All analyses were performed using GraphPad Prism software (Version 6.04). Results were considered significant at $p < 0.05$ (*$p < 0.05$, **$p < 0.01$, ***$p < 0.001$).

# Acknowledgements

The authors thank CGD patients and MPO-deficient patient for their participation in this study; Bärbel Raupach, Borko Amulic, CJ Harbort, Lorenz Knackstedt, Gabriel Sollberger, and Thea Tilley for their constructive comments on the manuscript. This work was supported by the Max Planck Society and in part by NIH grant GM118112.

# Additional information

### Competing interests

PRT: Consultant to Bristol-Myers Squibb. The other authors declare that no competing interests exist.

### Funding

| Funder | Grant reference number | Author |
| --- | --- | --- |
| Max-Planck-Gesellschaft | | Elaine F Kenny<br>Alf Herzig<br>Arturo Zychlinsky |
| National Institutes of Health | GM118112 | Aaron Muth<br>Santanu Mondal<br>Paul R Thompson |

The funders had no role in study design, data collection and interpretation, or the decision to submit the work for publication.

### Author contributions

EFK, Conceptualization, Data curation, Formal analysis, Investigation, Methodology, Writing—original draft, Writing—review and editing; AH, Data curation, Methodology, Writing—review and editing; RK, AM, PRT, HvB, Resources, Writing—review and editing; SM, Resources, Santanu Mondal was added as an author during the revision process as he synthesized the PAD inhibitors requested in reviewer comment 3. As the revision process was time sensitive he was very helpful in ensuring the inhibitors would be generated and provided to us in a timely manner that allowed us to answer point 3; VB, Resources, Data curation, Writing—review and editing; AZ, Conceptualization, Supervision, Funding acquisition, Writing—original draft, Writing—review and editing

## Author ORCIDs

Elaine F Kenny, http://orcid.org/0000-0001-9985-5620

Aaron Muth, http://orcid.org/0000-0002-0646-9964

## Ethics

Human subjects: Blood samples were collected according to the Declaration of Helsinki with study participants providing written informed consent. All samples were collected with approval from the ethics committee-Charité -Universitätsmedizin Berlin.

## Additional files

### Supplementary files

• Supplementary file 1. Graphical abstract: The diverse mechanisms of NETosis. In this study, we investigated whether NETosis occurs through a single signalling pathway or is induced by the five stimuli of interest in a diverse manner. As demonstrated, NETosis in response to *C. albicans* and GBS requires ROS, MPO and NE and induces histone H3 citrullination. This is in comparison to the NETosis seen in response to A23187 and nigericin during which none of the molecules highlighted above are required but citrullination of histone H3 does occur. Finally, we re-confirm that PMA-induced NETosis requires ROS, MPO and NE but does not result in the citrullination of histone H3. DOI: 10.7554/eLife.24437.030

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
