## [Decision Letter]

Thank you for submitting your article "Diverse Stimuli engage different Neutrophil Extracellular Trap pathways" for consideration by *eLife*. Your article has been favorably evaluated by Ivan Dikic (Senior Editor) and three reviewers, one of whom is a member of our Board of Reviewing Editors. The reviewers have opted to remain anonymous.

The reviewers have discussed the reviews with one another and the Reviewing Editor has drafted this decision to help you prepare a revised submission.

Summary:

Neutrophil extracellular traps appear to have several functions including defence against some bacterial and fungal pathogens. In the current manuscript, the authors investigated mechanisms involved in the formation of neutrophil extracellular traps. The authors stimulated neutrophils with 5 widely varied stimuli; the mitogen PMA, the ionophores A23187 and nigericin (bacterial toxin), the fungus *Candida albicans* and the gram positive bacteria Group B Streptococcus (GBS). According to their results, PMA, *C. albicans* and GBS use a related pathway for NET induction whereas ionophores require an alternative pathway. Thus, NETosis occurs through several signalling mechanisms, suggesting that extrusion of NETs is important in host defence. The analysis of signalling pathways also relied on the test of neutrophils from patients with deficiencies in pathways important for NET induction (CGD patients, and an MPO-deficient individual) along with primary healthy neutrophils treated with small molecule inhibitors of proteins required for PMA-induced NETosis. The authors also demonstrate that while apoptosis, necrosis and necroptosis can be induced in primary healthy neutrophils these pathways do not result in NET formation. Also inhibiting the induction of these cell death pathways has no effect on NETosis thus verifying that NETosis is a novel cell death pathway.

Essential revisions:

The reviewers found the work interesting and important. It clarifies that the pathways involved in NET formation vary depending on the stimulus to which neutrophils are exposed. However, at the same time some important questions arose which need to be carefully addressed:

1) One very interesting finding described in the manuscript is the effect of *C. albicans* on NET formation by CGD neutrophils. From their data, the authors conclude that *C. albicans* provided ROS for NET formation. They hypothesise that this might be the difference to *A. fumigatus* and explain why CGD patients are more susceptible to invasive aspergillosis rather than to a *C. albicans* infection. I think the data are suggestive for this conclusion. However, it would add strength if the authors had some experimental hints. It is most likely that ROS derive from the NADPH oxidase complex present in *C. albicans*. Possibly, the authors could carry out an experiment in which the NADPH oxidase is inhibited by DPI in *C. albicans* and such cells are used for the co-incubation experiment. Alternatively, an NADPH oxidase mutant of *C. albicans* could be used.

2) Figure 6: Why are there 2 bands for the A23 lane of citrullinated H3?

3) Depending on the donor, netting neutrophils appear to differ in their ability to downregulate NET formation in response to PAD inhibitors. It would be important to expand the PAD inhibitor experiments to about 10 healthy controls. In addition, it was found that difference sources of PAD inhibitors may differ in their ability to inhibit NETosis. As Cl^-^amidine has been shown by several groups to inhibit NETs in human neutrophils obtained from individuals with various inflammatory conditions as well as healthy controls (Smith CK A&R 2014; Subramaniam V J Med Chem 2015; Kusunoki Y. Front Immunol 2016; Rocha et al. Science Rep 2015; Hosseinzadeh et al. JLB 2016; Khandpur. ScienceTrans Med 2013), perhaps adding this compound to in vitro experiments would also be helpful.

4) A time course would be good. Especially at quite early times to see how quickly molecules like the ionophore release NETs.

5) Do all of these stimuli induce significant LDH release?

6) Do all of the stimuli induce the same quality of NETs? For example do they all have the same proteolytic activity?

7) Are all of the NETs equally capable of killing bacteria?

8) Are all of the NETs primarily from nuclear rather than mitochondrial material?

9) Include the demographics of the donors.

---

## [Author Response]

Essential revisions:

The reviewers found the work interesting and important. It clarifies that the pathways involved in NET formation vary depending on the stimulus to which neutrophils are exposed. However, at the same time some important questions arose which need to be carefully addressed:

1) One very interesting finding described in the manuscript is the effect of C. albicans on NET formation by CGD neutrophils. From their data, the authors conclude that C. albicans provided ROS for NET formation. They hypothesise that this might be the difference to A. fumigatus and explain why CGD patients are more susceptible to invasive aspergillosis rather than to a C. albicans infection. I think the data are suggestive for this conclusion. However, it would add strength if the authors had some experimental hints. It is most likely that ROS derive from the NADPH oxidase complex present in C. albicans. Possibly, the authors could carry out an experiment in which the NADPH oxidase is inhibited by DPI in C. albicans and such cells are used for the co-incubation experiment. Alternatively, an NADPH oxidase mutant of C. albicans could be used.

The reviewer raises an interesting point, to address it we treated the *C. albicans* with 10 µM pyrocatechol during the 30 minute opsonisation step prior to NET induction. This new data is shown in Figure 3—figure supplement 1 and described in the fourth paragraph of the subsection “Differential ROS requirements of NETs”. It revealed that blocking ROS in *C. albicans* also reduced the amount of NETs generated by healthy neutrophils. Moreover, pre-treating the neutrophils with pyrocatechol and then adding *C. albicans* that were pre-treated with the ROS scavenger as well resulted in even greater NET inhibition. This demonstrates that fungus generated ROS can contribute to make NETs.

Interestingly, it was recently demonstrated that CGD patients produce significantly less NETs in response to *A. fumigatus* than to *C. albicans* (Gazendam et al., 2016). This data addresses the observation of the reviewer on the susceptibility to *A. fumigatus* and *C. albicans* in CGD patients and is now discussed in the fifth paragraph of the Discussion.

2) Figure 6: Why are there 2 bands for the A23 lane of citrullinated H3?

The reviewer correctly points out that this blot is confusing to the reader. Therefore, we included new data showing that during the process of NETosis histone H3 is processed by granule proteases. In the new version of the manuscript we show (Figure 7 and subsection “All stimuli induce NETs that are proteolytically active, kill bacteria and are composed primarily of nuclear DNA”, first paragraph) that histone H3 is cleaved 90 minutes post stimulation in response to all stimuli. These data show that the two bands seen in response to A23187 stimulation are the result of histone cleavage.

3) Depending on the donor, netting neutrophils appear to differ in their ability to downregulate NET formation in response to PAD inhibitors. It would be important to expand the PAD inhibitor experiments to about 10 healthy controls. In addition, it was found that difference sources of PAD inhibitors may differ in their ability to inhibit NETosis. As Cl^-^amidine has been shown by several groups to inhibit NETs in human neutrophils obtained from individuals with various inflammatory conditions as well as healthy controls (Smith CK A&R 2014; Subramaniam V J Med Chem 2015; Kusunoki Y. Front Immunol 2016; Rocha et al. Science Rep 2015; Hosseinzadeh et al. JLB 2016; Khandpur. ScienceTrans Med 2013), perhaps adding this compound to in vitro experiments would also be helpful.

The reviewer correctly points out a discrepancy in the field and we followed his/her suggestion. To address this comment we tested the three PAD inhibitors on neutrophils from 10 independent healthy donors. As shown in Figure 6 and described in the second paragraph of the subsection “Histone citrullination occurs but is not required for NET induction”, pre-treatment of the cells with the Cl^-^amidine PAD inhibitor had no effect on NETosis. This was also true for cells pre-treated with BB-Cl^-^amidine (Figure 6) and TDFA (Figure 6). The data shown in Figure 6 is the mean ± SEM of the 10 independent experiments. These experiments are also shown individually in Figure 6—figure supplement 2 which confirm that these inhibitors do not affect NET induction in response to any of the stimuli tested.

As an important control, we also included data showing that the Cl^-^amidine blocks citrullination of histone H3 in response to A23187, *C. albicans* and GBS stimulation (Figure 6—figure supplement 1). This demonstrates that the compound was active under our experimental conditions.

4) A time course would be good. Especially at quite early times to see how quickly molecules like the ionophore release NETs.

This is a helpful suggestion by the reviewer and the data are now included in Figure 1—figure supplement 1 and are described in the second paragraph of the subsection “A wide range of stimuli induce Neutrophil Extracellular Traps (NETs)”. These data show that all 5 stimuli tested generate NETs over a 3-4 hour time course.

5) Do all of these stimuli induce significant LDH release?

We addressed this relevant question and show in Figure 8—figure supplement 1 that all five stimuli induce LDH release in the process of making NETs. This is described in the second paragraph of the subsection “NETosis is distinct from other forms of cell death”.

6) Do all of the stimuli induce the same quality of NETs? For example do they all have the same proteolytic activity?

The reviewer raises interesting questions about the quality of the NETs in this and the next two questions. To address proteolytic activity we examined the ability of neutrophils to degrade histone H3 during NETosis and show that all 5 stimuli lead to degradation within 90 minutes as shown in Figure 7. We also isolated the NETs produced in response to the 5 stimuli and tested their ability to breakdown FITC-labelled casein using the Pierce Fluorescent Protease Assay Kit (Thermo scientific) as directed in the manufacturer’s instructions. This data are shown in Figure 7 and described in the first paragraph of the subsection “All stimuli induce NETs that are proteolytically active, kill bacteria and are composed primarily of nuclear DNA”.

7) Are all of the NETs equally capable of killing bacteria?

To address this question we include a new Figure 7 which is described in the second paragraph of the subsection “All stimuli induce NETs that are proteolytically active, kill bacteria and are composed primarily of nuclear DNA”. In this experiment we induced NETs and 4 hours later treated the neutrophils with Cytochalasin D to inhibit phagocytosis. We included samples treated with DNase 1 at 50U/ml to degrade the NETs. Tetracycline resistant *E. coli* (XL1-Blue, Stratagene. Used to differentiate between the GBS, *C. albicans* and *E. coli* growth on agar plates) at a MOI of 1 was added to the cells/NETs and incubated at 37°C for 1 hour. The cells and *E. coli* were then scrapped and collected. One set of samples were sonicated to release any bacteria trapped by the NETs and unavailable for counting. Finally serial dilutions were carried out and the *E. coli* were plated on agar plates containing tetracycline, incubated at 37°C for 24 hr before counting. The data showed that all NETs produced killed *E. coli*.

8) Are all of the NETs primarily from nuclear rather than mitochondrial material?

In the new version of the manuscript, we include Figure 7 and described the results in the third paragraph of the subsection “All stimuli induce NETs that are proteolytically active, kill bacteria and are composed primarily of nuclear DNA”. This figure shows that all NETs generated are primarily composed of nuclear DNA.

9) Include the demographics of the donors.

The demographics of the CGD patients has been included in Table 1 statement on the source of the healthy blood donors stating that blood was anonymously collected at the Charité hospital, Berlin has also been included in the Materials and methods subsection “Donor consent”.